# Knowledge and practices of exclusive breastfeeding among mothers in rural areas of Rajshahi district in Bangladesh: A community clinic based study

**Md. Masud Rana[1], Md. Rafiqul Islam[1]\*, Md. Reazul Karim[1], Ahmed Zohirul Islam[1], Md. Akramul Haque[2], Md. Shahiduzzaman[3], Md. Golam Hossain[4]**

1 Department of Population Science and Human Resource Development, University of Rajshahi, Rajshahi, Bangladesh, 2 DASCOH Foundation, Rajshahi, Bangladesh, 3 Department of Electrical and Electronic Engineering, Northern University, Dhaka, Bangladesh, 4 Department of Statistics, University of Rajshahi, Rajshahi, Bangladesh

\* rafique_pops@yahoo.com

## Abstract

### Background

World Health Organization (WHO) suggests that exclusive breastfeeding (EBF) is the best nutrition for the neonate. Still, it remains a big challenge to establish EBF not only in Bangladesh but also in any developing countries.

### Objective

The aim of this study was to determine the level of knowledge and practices on EBF and its relationship between different socioeconomic and demographic factors among mothers having at least one child of aged 6–12 months in the rural area of Rajshahi District, Bangladesh.

### Methodology

A community clinic (CC) based study has been conducted by using semi-structured questionnaire. A total of 513 mothers having at least one child aged 6–12 months from 32 different CC in the rural area of Rajshahi District, Bangladesh during September to December 2015. A composite index, chi-square test, and logistic regression model were utilized in this study.

### Results

The prevalence of knowledge and practices on EBF were 34.5% and 27.9% among mothers having at least one child aged 6–12 months. From the analyses, mothers age of ≥21 years were (adjusted odds ratio (AOR) = 13.840, 95% CI: 7.394–25.904; p<0.001) times more likely to have knowledge on EBF and (AOR = 0.084, 95% CI: 0.050–0.143; p<0.05) times less likely to have practices of EBF compared to mother's age ≤20 years. Service holders

**Data Availability Statement:** All relevant data are within the manuscript and Supporting Information files.

**Funding:** The author(s) received no specific funding for this work.

**Competing interests:** The authors declare that they have no competing interest.

mothers were (AOR = 9.992, 95% CI: 4.485–22.260, p<0.05) times more likely to have practices than that of house wife. Home delivery mothers was (AOR = 0.208, 95% CI: 0.111–0.389; p<0.05) times less likely to have practices of EBF than that of the hospital delivery mothers. Those mothers monthly family income ≥10,000 Bangladeshi taka (BDT) currency was (AOR = 0.092, 95% CI: 0.050–0.168, p<0.05) times less likely to have practices of EBF compared to their counterparts.

## Conclusions

This study was found poor knowledge and practices on EBF. This study suggested that education and EBF related intervention could play an important role to increase good knowledge and practices on EBF among mothers. Malnutrition will be decreased if EBF was widely established in Bangladesh.

## Introduction

Durable evidence specifies that exclusive breastfeeding (EBF) is one of the best nutrition practices for children health, growth and nutrition and termed as an optimal strategy for feeding newborn and young infants [1–5]. According to WHO and UNICEF, EBF should start within less than one hour of delivery and must have to continue up to 6 months of infant's age as it is the only source diets or fluids for babies at that age and have to sustain with balancing feeding on at the minimum 24 months of infants age [6]. Malnutrition in all its forms, either indirectly or directly, is responsible for about half of the all deaths including infants worldwide [7]. Children especially new born babies are at large danger of malnutrition from first six months of life when breast milk alone is necessary to meet all nutritious supplies and balancing feeding needs to be in progress [8, 9]. Good practice of EBF can prevent 13.8% of all deaths among infants aged less 2 years and 11.6% of under 5-years children deaths [10–12] but unfortunately a report estimated in 2012 that, only 35% of infants were exclusively breastfed globally [13]. EBF, by its various recognized health welfare for babies, children, and their mothers, was a crucial plan to recover public health [14]. Low breastfeeding rates were found in Canada, and especially in industrialized countries [15]. Besides, EBF at least 6 month's was not a general practices in advanced nations, and it was still fewer in developing nations [16]. The key gap was sympathetic to the physical, organic, cultural and socio economic factors of the length of EBF. In this review, nutrient capability of EBF is maximum usually assessed for infant development [17]. In the South East Asia, it was reported that prevalence of EBF were 55.0%, 66.1%, 51.4%, 70.0% for India, Nepal, Bhutan, Sri Lanka respectively [18–21].

In global health congress, EBF increased to ensuing first six months [22–24]. Nearly Bangladeshi children are breastfed to an approximate extent to the initial year of life and luckily, the continuance of breastfeeding is up to the second year of baby's life, through 91% ongoing to breastfeed [25]. Bangladesh has a key amounts of malnourishment in South East Asia with nearly partial for children aged 59 months being underweight and 64%, 36%, 35.90% EBF reported [23,24,26,27]. To determine the poor newborn nourishing knowledge, practices and it doubles of the exact aims of the state plan for baby and early born children feeding [28].

Several studies have been performed to assess the knowledge, perception and practices on breastfeeding among females and shown global trends in EBF [13, 28], knowledge, attitude and techniques of breastfeeding under five children mothers in Nigerian [29,30], knowledge and practices of EBF in Nigerian populations [28]. Special concern has been paid to the

association between school going girl's perception and knowledge about breastfeeding [31], knowledge, and practice of baby's moms towards EBF and its related issues in Ethiopia [32–34]. Though only a few studies have been carried out in this regard, most of these studies were carried out in developed country settings [35–36]. Furthermore, concerning issued the measurement of knowledge and practices about EBF have not been adequately addressed in earlier studies. The difficulty of judging knowledge lies in its multidimensional aspects. Most of the studies have been focused on a few indicators. In Bangladesh most of the rural village covered by the community clinic so, this study measured by composite index. Therefore, the aim of this study was to assess the knowledge and practices on EBF among mothers who having at least one child aged 6–12 months in Rajshahi District, Bangladesh.

## Methods

### Design and study population

A community clinic based study was conducted in the rural area of Rajshahi district, Bangladesh. There are several reasons why we select the mothers having at least one child aged 6–12 months from different CCs in Rajshahi district. This area is situated at the nearest of Rajshahi is identified as divisional and education city [37]. Most of the subjects were living at the different catchment area of CCs in Paba upazila (sub-district) Rajshahi district, Bangladesh.

### Simple size determination

Since study population is well-known mothers having at least one child age 6–12 months, the following formula has been used for collecting sample size: $n = N/ (1+Nd^2)$, where n = required sample size, N = population size (5,123), d = marginal error (0.05). The formula provided that the minimum sample size was estimated to be 366 for this study. But, for better results, we have collected data from 513 respondents [38].

Before sampling, 6–12 month children lists were got from the Paba upazila Health Complex, Rajshahi that list used for expanded programme on immunization. A two-stage purposive sampling approach was accepted to choose the mothers having at least one child age 6–12 months from Rajshahi district. In the first stage, out of nine upazila of Rajshahi district, purposively selected one upazila named Paba. In the second stage, purposive sampling technique was used to select sample size. There were 232 CCs in Rajshahi district and 32 CCs in Paba upazala. In Paba upazila, CCs catchment area, 5123 mothers were living who were taken different types of general, maternal, child health, family planning and expended program on immunization (EPI) services from CCs. In Paba, 530 mothers were selected out of 5,123, mothers who having at least one child age 6–12 months and finally 513 samples were successfully interviewed [39].

### Data collection

In this study, data were collected from September to December'2015 (i) socio-demographic characteristics and (ii) knowledge about EBF using a semi-structured questionnaire. The questionnaire was reviewed by five specialists and assistants, and initially piloted to validate the questionnaire. The questionnaire also modified based on piloted results of the pre-test exercise to make it easier to realize and for response. Five fully trained and experienced enumerators conducted the interview.

### Outcomes variables

The dependent variable in this study is the level of knowledge about EBF, which was measured through nine different questions, namely: i) What do you mean by EBF? ii) Do you know

when EBF should be started? iii) Do you know when supplementary feeding is needed? iv) Do you know water should allow in EBF period? v) Do you know honey should allow in EBF period? vi) Do you know what the appropriate duration of EBF is? vii) Do you know what the benefits of EBF? viii) Do you know what is happen if EBF were not done? ix) Do you know any additional feed is essential during EBF period?

Another outcome variable in this study was practices about EBF, which was measured through two different questions, namely: i) Do you feed any complement food for your last baby during EBF? ii) What type of feed you allow during your EBF period for your last baby? The respondent's knowledge and practice were scored used a system adopted from earlier studies. The individually appropriate reply was given 1 point, while improper replies received 0 point [40].

## Independent variables

In this study, socio-economic and demographic factors were included as independent variables. Age classified into two groups: a group ($\leq 20$ years), and other age groups ($\geq 21$ years). Place of delivery was divided into two group hospital and home and occupation also classified into two categorized such as housewife and service holder. Education was classified based on the formal learning system in Bangladesh: Illiterate (0 years), primary education (1–5 years), secondary and higher (6 years or more). Type of family was categorized as per joint or single family. Respondents monthly income was categorized as $\leq$9,999 Bangladeshi taka (BDT) currency or $\geq$10,000 Bangladeshi taka (BDT) currency considered as yes or no.

## Statistical analyses

Statistical Package for Social Science (SPSS) version 22 IBM was used for analyzing the data. Descriptive analyses were conducted to ascertain the socio-economic and demographic factors by knowledge and practices of EBF of the subjects. Socio-economic and demographic differences regarding knowledge and practices of EBF were assessed by $\chi^2$ test through significance for all analyses was set at p<0.05. The study designed was completely adjusted models to analyze each binary outcome variables. This research entered all the variables instantaneously into the binary logistic regression models. The adjusted odds ratio (AOR) was observed to assess the strength of the association's at 95% CI for significance test.

The knowledge index was created through the sums of binary input variables, where the highest and lowest values were selected for each underlying pointer. The enactment of individually pointer was articulated using a unit-free index between 0 and 1 in accordance with the structure technique of the Human Development Index [41].

Knowledge index = (Actual value- Minimum value) / (Maximum value-Minimum value)

The scores was created then categorized as the groups labeled as poor and good knowledge and practices [41, 42]. Before measuring the level of knowledge and practices using consistent or reliability of respondents answer. The knowledge score is poor = 0–4 and good = 5–9 and the practice level score is poor = 0 <1, good = $\geq$1 [42].

## Ethics approval and consent to participate

CCs based activities implemented by Development Association for Self-reliance Communication and Health (DASCOH Foundation) and financially supported by Swiss Red Cross (SRC). DASCOH Foundation was got approval from Community Based Health Care (CBHC) under the Ministry of Health and Family Welfare Government of Bangladesh. Ethics committee of Non-governmental Affairs Bureau (NGOAB) of Bangladesh has approved this project. Before

**Table 1. Socio-demographic characteristics according to knowledge on EBF.**

| Characteristics | Knowledge about EBF | | | |
|---|---|---|---|---|
| | N (%) | Good | Poor | p-value |
| | | 177 (34.5%) | 336 (65.5%) | |
| **Age in years** | | | | 0.001 |
| ≤20 years | 314(61.2) | 164 (52.2) | 150(47.8) | |
| ≥21 years | 199(38.8) | 13(5.5) | 186(93.5) | |
| **Religion** | | | | 0.001 |
| Muslim | 408(79.5) | 115(28.2) | 293(71.8) | |
| Non-Muslim | 105(20.5) | 62(59.0) | 43(41.0) | |
| **Delivery place** | | | | 0.112 |
| Hospital | 309(60.2) | 115(37.2) | 194(62.8) | |
| Home | 204(39.8) | 62(30.4) | 142(69.6) | |
| **Occupation** | | | | 0.370 |
| Housewife | 308(60.0) | 111(36.0) | 197(64.0) | |
| Service holder | 205(40.0) | 66(32.2) | 139(67.8) | |
| **Educational status** | | | | 0.108 |
| Illiterate | 141(27.5) | 56(39.7) | 85(60.3) | |
| Primary | 98(19.1) | 26(26.5) | 72(73.5) | |
| Secondary & Higher | 274(53.4) | 95(34.7) | 179(65.3) | |
| **Type of family** | | | | 0.001 |
| Joint | 408(79.5) | 115(28.2) | 293(71.8) | |
| Single | 105(20.5) | 62(59.0) | 43(41.0) | |
| **Monthly family income in BDT** | | | | 0.126 |
| ≤9,999 | 307(59.8) | 114(37.1) | 193(62.9) | |
| ≥10,000 | 206(40.2) | 63(30.6) | 143(69.4) | |

going to data collection this study were shortly discussed with the participants about its objectives. For this study researchers were received written consent from all of the subjects.

## Results

In the study, it was surveyed that the knowledge and practices on EBF among mothers. Table 1 showed the socio-economic and demographic factors about knowledge on EBF related appearances of the respondents. A total of 513 individuals were involved in this study. From the total sample population, approximately 61.2% and 79.5% were ≤ 20 years of age and Muslim. Out of 513 simple, 79.5% of respondents came from joint family. Furthermore, it was found that these characteristics were statistically significant (p<0.05). 34.5% respondents had a good knowledge on EBF and here delivery place, education and monthly family income were statistically insignificant with knowledge level.

Table 2 showed the socio-economic and demographic factors about practices on EBF related appearances of the respondents. A total of 513 individuals were involved in this stud. From the total sample population, 61.2% and 78.9% were ≤ 20 years of age and Muslim group respectively. Out of 513 simple, 60.2% of deliveries were at hospital and 61.4% respondents were housewife. In case of education, 27.5% respondents were illiterate, 19.1% were primary educated and the remaining 53.4% had secondary or higher level of education and 79.5% of respondents came from joint family. A major portion of respondent's (59.8%) monthly family income was below 9,999 BDT. From the total sample population, 27.9% respondents have

**Table 2. Socio-demographic characteristics according to EBF practices.**

| Characteristics | Practices on EBF | | | |
|---|---|---|---|---|
| | N (%) | Good | Poor | p-value |
| | | 143(27.9%) | 370(72.1%) | |
| **Age in years** | | | | 0.001 |
| ≤20 years | 314(61.2) | 122(38.9) | 192(61.1) | |
| ≥21 years | 199(38.8) | 21(10.6) | 178(89.4) | |
| **Religion** | | | | 0.001 |
| Muslim | 405(78.9) | 143(35.3) | 265(64.7) | |
| Non-Muslim | 108(21.1) | 0(0), | 108(100.0) | |
| **Delivery place** | | | | 0.001 |
| Hospital | 309(60.2) | 130(42.1) | 179(57.9) | |
| Home | 204(39.8) | 13(6.4) | 191(93.6) | |
| **Occupation** | | | | 0.001 |
| Housewife | 315(61.4) | 136(43.2) | 179(56.8) | |
| Service holder | 198(38.6) | 07(3.5) | 191(96.5) | |
| **Educational status** | | | | 0.006 |
| Illiterate | 141(27.5) | 89(63.1) | 52(36.9) | |
| Primary | 98(19.1) | 80(81.6) | 18(18.4) | |
| Secondary & Higher | 274(53.4) | 201(73.4) | 73(26.6) | |
| **Type of family** | | | | 0.001 |
| Joint | 408(79.5) | 143(35.1) | 265(64.9) | |
| Single | 105(20.5) | 0(0) | 105(100.0) | |
| **Monthly family income in BDT** | | | | 0.001 |
| <9,999 | 307(59.8) | 130(42.1) | 177(57.7) | |
| ≥10,000 | 206(40.2) | 13(6.3) | 193(93.7) | |

practice on EBF. Furthermore, it was found that these characteristics were statistically significant ($p < 0.05$).

It was found that the model chi-square was 134.104 (p-value = 0.001) and Nagelkerke $R^2$ of the fitted model was 0.318 which expressed the good fit of the model that were shown at the bottom of Table 3. Regression analysis of the factors associated with the knowledge and among

**Table 3. Effects of socio-economic and demographic factors for knowledge on EBF.**

| Characteristic | p-value | Adjusted odds ratio (AOR) | 95% CI | |
|---|---|---|---|---|
| | | | Lower | Upper |
| **Age in years** | | | | |
| ≤20 years [R] | | | | |
| ≥21 years | 0.001 | 13.840 | 7.394 | 25.904 |
| **Religion** | | | | |
| Muslim [R] | | | | |
| Non-Muslim | 0.358 | 4.607 | 0.177 | 119.916 |
| **Type of family** | | | | |
| Joint [R] | | | | |
| Single | 0.510 | 3.004 | 0.114 | 79.057 |

Model summary:

Model chi-square = 134.104 (p-value = 0.001), Nagelkerke $R^2$ of the fitted model = 0.318.

**Table 4. Effects of socio-economic and demographic factors for practices on EBF.**

| | p-value | AOR | 95% C.I | |
|---|---|---|---|---|
| | | | Lower | Upper |
| **Age in years** | | | | |
| ≤20 years [R] | | | | |
| ≥21 years | 0.001 | 0.084 | 0.050 | 0.143 |
| **Religion** | | | | |
| Muslim [R] | | | | |
| Non-Muslim | 0.057 | 0.061 | 0.003 | 1.092 |
| **Delivery place** | | | | |
| Hospital [R] | | | | |
| Home | 0.001 | 0.208 | 0.111 | 0.389 |
| **Occupation** | | | | |
| Housewife [R] | | | | |
| Service holder | 0.001 | 9.992 | 4.485 | 22.260 |
| **Educational status** | | | | |
| Illiterate [R] | | | | |
| Primary | 0.752 | 1.192 | 0.401 | 3.547 |
| Secondary & Higher | 0.419 | 1.435 | 0.597 | 3.452 |
| **Type of family** | | | | |
| Joint [R] | | | | |
| Single | 0.518 | 2.500 | 0.155 | 40.304 |
| **Monthly family income in BDT** | | | | |
| ≤9,999 [R] | | | | |
| ≥10,000 | 0.001 | 0.092 | 0.050 | 0.168 |

Model summary:

Model chi-square = 388.475 (p-value = 0.001) and Nagelkerke $R^2$ of the fitted model = 0.765.

mothers on EBF. Children mothers age ≥21 years (adjusted odds ratio, (AOR) = 13.840; 95% confidence interval CI: 7.394–25.904), mothers were more likely to have a good knowledge on EBF compared to their counterparts and it was statistically significant (p<0.05).

In Table 4, model summary of the level of practices on EBF was also demonstrated at the bottom of this table. It was found that the model chi-square was 388.475 (p value = 0.001) and Nagelkerke $R^2$ of the fitted model was 0.765 which also express the good fit of the model. The respondents aged ≥ 21years (AOR = 0.084; 95% CI: 0.050–0.143), home (AOR = 0.208; 95% CI: 0.111–0.389), ≥ 10,000 BDT monthly family income (AOR = 0.092; 95% CI: 0.050–0.168) children mothers were less likely to have practices on EBF compared to their counterparts and these factors were statistically significant (p<0.05). Moreover, service holder (AOR = 9.992; 95% CI: 4.485–22.260) mothers were more likely to have good practices on EBF than their counterparts and it was statistically significant (p<0.05).

## Discussion

This study surveyed the knowledge and practices on EBF among mothers at rural area of the CCs in Rajshahi district, Bangladesh. It were observed that prevalence of EBF were Eastern and Southern Africa 55%, South Asia 54%, Latin America and the Caribbean 38%, North America 35%,West and Central Africa 34%, Eastern Europe and Central Asia 33%, East Asia and Pacific 30%, Middle East and North Africa 30% and Globally 42% [26]. This study

assumed that, as the area was situated at very near of divisional and educational city and that most of the mothers living in these area of CCs, maximum respondents were knowledgeable and practicable on EBF, but this study demonstrated that a few percent of mothers in this area have knowledgeable and practicable on EBF. In Bangladesh similar study were found in practice level on EBF [43, 44]. The study found that, younger respondents' $\leq 20$ years had knowledge and practices as likened to the older respondents' $\geq 21$ years similar results were found in others countries [45]. It was found that hospital delivery respondents had practices as associated to their counterparts. Study results was consistent with previous other study in Ethiopia [46, 47]. An extra assumption was that most of the house wife mother, secondary and higher level of educated mothers knew EBF benefit so their practices level is high as paralleled with the housewife respondents. We found that housewife respondents have practices as matched to their counterpart. A picture of rural area housewife respondents watched Bangladesh television. This study result is consistent with previous other studies [48, 49]. This study was assumption that educated mothers have and practices. Joint family member's mothers have knowledge and practices were different as compared to the single family mothers. This study observed that Joint family member's mothers in our study share their findings with other family members but not properly follow various types of religious orthodoxy. Though, study highlights the need for EBF health education program educations children mothers. It was found that $\leq 9,999$ BDT income respondents have practice as compared to their counterparts. Similar results were found in previous other studies [50, 51]. An extra assumption that most of the low monthly income mother watched BTV and follow those indications as compared with the rich housewife mothers they watch different type of Indian serial through dish channel [49].

There are major two findings. First, study have found very poor knowledge and practices 34.50%, 27.9% on EBF was noted among study participants. Second, mothers have knowledge and practice about EBF who were aged $\leq 20$ years, Housewife, hospital delivery, joint family members and $\leq 9,999$ BDT monthly family income mothers were more likely to have knowledge and practices on EBF.

As a final point, the idea of knowledge and practices on EBF, which has some definitions; so, it is challenging to measure, particularly using the questionnaire. However, this study measures the knowledge and practices through a lot of indicators which were reflected by several previous studies [26,52–55].

This study had some limitations such as it was a community clinic based study so, it did not permit us to create any complete progressive associations for identifying knowledge and practices on EBF and several socio-economic and demographic factors and health performance linked with this features. This study suggested for more longitudinal research to realize this complex relationship and understand the fundamental tools. Secondly, there are 64 districts and 491 sub-districts (upazilas) in Bangladesh, in this study, considered only one district and one upazila of considering district.

## Conclusions

This study found poor knowledge and practices on EBF among mothers. Binary logistic regression model demonstrate that a few number of factors were influencing EBF knowledge and practices. Female education should be increased substantially and hence more job opportunities could be created in various dimensional job sectors for them. This study also suggested that social safety net program could play an important role to increase knowledge and practices on EBF among mothers. Malnutrition will be decreased if EBF was widely established. This study provides important information to improving knowledge and practices on EBF in removing malnutrition from Bangladesh.

## Supporting information

**S1 File.**
(SAV)

## Acknowledgments

The authors gratefully acknowledge the authority of CCs of Rajshahi District, Bangladesh for giving permission to take data from CCs catchment area. We also acknowledge Md. Ektier Uddin, DASCOH Foundation and Swiss Red Cross to help for data collection from CCs.

## Author Contributions

**Conceptualization:** Md. Masud Rana, Md. Rafiqul Islam.

**Data curation:** Ahmed Zohirul Islam, Md. Shahiduzzaman.

**Formal analysis:** Md. Masud Rana.

**Methodology:** Md. Reazul Karim.

**Validation:** Md. Akramul Haque.

**Visualization:** Md. Shahiduzzaman.

**Writing – original draft:** Md. Masud Rana.

**Writing – review & editing:** Md. Rafiqul Islam, Md. Golam Hossain.

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
