## [Decision Letter · Decision Letter 0]

19 Dec 2019

PONE-D-19-28733

Knowledge and practice of exclusive breastfeeding among mothers in rural areas of Rajshahi district in Bangladesh: A community clinic based study

PLOS ONE

Dear Md. Islam,

Thank you for submitting your manuscript to PLOS ONE. After careful consideration, we feel that it has merit but does not fully meet PLOS ONE’s publication criteria as it currently stands. Therefore, we invite you to submit a revised version of the manuscript that addresses the points raised during the review process.

We would appreciate receiving your revised manuscript by 26 January 2020. To enhance the reproducibility of your results, we recommend that if applicable you deposit your laboratory protocols in protocols.io, where a protocol can be assigned its own identifier (DOI) such that it can be cited independently in the future. For instructions see: http://journals.plos.org/plosone/s/submission-guidelines#loc-laboratory-protocols

We look forward to receiving your revised manuscript.

Kind regards,

Russell Kabir, PhD

Academic Editor

PLOS ONE

Journal Requirements:

2. Please include additional information regarding the survey or questionnaire used in the study and ensure that you have provided sufficient details that others could replicate the analyses. For instance, if you developed a questionnaire as part of this study and it is not under a copyright more restrictive than CC-BY, please include a copy, in both the original language and English, as Supporting Information. In the text, you refer to pre-testing of this questionnaire. Please provide detail concerning the number of participants and where they were recruited from.

3. Thank you for including your competing interests statement; "none"

We note that one or more of the authors are employed by a commercial company: DASCOH Foundation

4. We note that [Figure(s) 1] in your submission contain [map/satellite] images which may be copyrighted. All PLOS content is published under the Creative Commons Attribution License (CC BY 4.0), which means that the manuscript, images, and Supporting Information files will be freely available online, and any third party is permitted to access, download, copy, distribute, and use these materials in any way, even commercially, with proper attribution. For these reasons, we cannot publish previously copyrighted maps or satellite images created using proprietary data, such as Google software (Google Maps, Street View, and Earth). For more information, see our copyright guidelines: http://journals.plos.org/plosone/s/licenses-and-copyright.

1.    You may seek permission from the original copyright holder of Figure(s) [1] to publish the content specifically under the CC BY 4.0 license. 

6. Please amend your list of authors on the manuscript to ensure that each author is linked to an affiliation. Authors’ affiliations should reflect the institution where the work was done (if authors moved subsequently, you can also list the new affiliation stating “current affiliation:….” as necessary).

7. Please ensure that you refer to Figure 1 in your text as, if accepted, production will need this reference to link the reader to the figure.

8. We note you have included a table to which you do not refer in the text of your manuscript. Please ensure that you refer to Table 3 in your text; if accepted, production will need this reference to link the reader to the Table.

Reviewers' comments:

Reviewer's Responses to Questions

**Comments to the Author**

1. Is the manuscript technically sound, and do the data support the conclusions?

Reviewer #1: Partly

Reviewer #2: Yes

2. Has the statistical analysis been performed appropriately and rigorously? 

Reviewer #1: Yes

Reviewer #2: Yes

3. Have the authors made all data underlying the findings in their manuscript fully available?

Reviewer #1: Yes

Reviewer #2: Yes

4. Is the manuscript presented in an intelligible fashion and written in standard English?

Reviewer #1: No

Reviewer #2: Yes

5. Review Comments to the Author

Reviewer #1: Comments to the Author

In this manuscript, the authors investigate the knowledge and practices on exclusive breastfeeding (EBF) and its relationship between different socioeconomic and demographic factors among mothers having at least one child (6-12 months age) in a rural area of Rajshahi District, Bangladesh. This study community clinic study with semi-structured questionnaires with sample size of 513 mothers. This study reports poor knowledge and practices on EBF, education, and EBF related intervention have an essential role in increasing knowledge and practices on EBF. Moreover, authors suggested malnutrition may be decreased if EBF was established in Bangladesh.

However, there are some suggestions and major concerns that must be addressed.

1. BMC Pediatr. 2018; 18: 93 (PMID: 29499670) Previous publication from the same group with similar kind of study (a country based cross-sectional study including Rajshahi) and its outcome similar; What is the novelty in this study? In what means this study “best of my knowledge none study has not been conducted on EBF” Line 108 statement?

2. Whether data used in the present study was extracted from the large scale of data-set collected by Bangladesh Demographic and Health Survey (BDHS) -2014? Although, Data collection period mentioned September to December 2015, is that part of BDHS? For clarification explanation must be included.

3. Line: 196 “Monthly family income was below 9,999 BDT” What was the average monthly family income to predict the socioeconomic status of well-being as per statistic; references must be included.

4. Abbreviate at first instance, check throughout the manuscript. Example: adjusted odds ratio (AOR), Bangladeshi taka (BDT) currency (check-in abstract also).

5. Avoid repeated discussion/results and check for those portions.

6. Fewer graphical representations of relevant data (key study outcome) would be interesting along with table.

7. It is interesting to compare previously published studies, for example, “Prevalence of exclusive breastfeeding and associated factors among mothers in rural Bangladesh: a cross-sectional study” Int Breastfeed J 9, 7 (2014) DOI:10.1186/1746-4358-9-7.

8. There are many other minor errors of syntax and grammar throughout the text, which need to be fixed.

Reviewer #2: 1. The introduction section need more clarity esspecially the studies of Banglash need to be reffered for contexual understanding of the issues. Further, the studies of south east aia can be highlighted to understand the standng of breat feeding of bangladesh with the regional practices.

2. the methodology seems to not explained proporly sothat the tudy can be dulpicated elsewhere involveig the methods . As of now the methodology is scant and processes need to be explained.

3. The important finding sneed to be highlighted. there are many commion and generalizaed indings those need to be edit out to make the article crispy and easy to understand.

4. The disscussion section needs a lot of effort to bring the global perspective in the study and need high level of comparision. the factors like religion social groups and the practice of breast feeding need to be ompared.

5. the conclusion must be from the finding like breast feeding and the factors relating to practices of bangladesh. Make it strong.

6. PLOS authors have the option to publish the peer review history of their article (what does this mean?). If published, this will include your full peer review and any attached files.

Reviewer #1: No

Reviewer #2: Yes: Ranjit Kumar Dehury

---

## [Author Response · Author response to Decision Letter 0]

31 Mar 2020

Point wise response to academic editor comments: 

1. I ensure that our manuscript meets PLOS ONE‘s style.

2. We uploaded questionnaire of 

i) Original/Mother’s language.

ii) English language as supporting information.

3. The authors declare that they have no competing interest.

DASCOH Foundation is nonprofit organization and they have no any fund for publication and its supporting document is provided

4. Figure 1 is omitted from the manuscript

5. SPSS data file is uploaded in Data Availability statement 

6. We have already ensured the affiliation of each author

7. It is already deleted 

8. Table 3 is inserted in the respective section in the text of the manuscript (in the line of 206 in Manuscript file)

# 1. Reviewer comments:

1. BMC Pediatric. 2018; 18:93 is already studied. Some variable are same and some are different but the outcome variable is converted into composite index, then it is classified into two groups as good and poor.

“Dependent variable is converted to composite index”- This study worked on EBF in Rajshahi district as well as community clinic based study. Most of the researchers studied on EBF practices but this study worked about knowledge and practices on EBF. One of the novelty of this study was to introduce the composite index on EBF. “To the best of our knowledge none study has been conducted on EBF” this section is deleted from the manuscript

2. Actually the data of this study is primary collected during the period of September to December 2015. Consequently this data are not extracted from Bangladesh Demographic and Health Survey (BDHS) 2014. In fact, this study is not part of BDHS 2014.

3. According to World Bank national accounts data, and OECD National Accounts data files’2015 per capital income in Bangladesh 1,248.453 US dollar its exchange rate for December 31, 2015, 1 US dollar = 78.9031 BDT. For this reason we divided monthly family income into two categories such as ≤ 9,999 BDT and ≥ 10,000 BDT. 

4. Adjusted odds ratio (AOR) and Bangladeshi taka (BDT) is corrected in the abstract as well as in the respective sections (in the line 43 and 49 of Manuscript).

5. We carefully check and revise the respective portions of discussion or results. 

6. Since relevant information are giving in the respective table, so we did not provide figures.

7. We have compared our finding to the previous mentioned article name International Breastfeeding Journal 9,7(2.014) in the Introduction section of Manuscript

8. We have carefully checked the whole manuscript thoroughly.

# 2. Reviewer Comments: 

1. The studies of Bangladesh as well as South East Asia on EBF are added in the line of 79-81. 

2. Methodology section was revised thoroughly for proper explanation and understanding.

3. The article was rewriting for making article crispy and easy to understand that were indicated in the Results section

4. The discussion section was edited minutely, carefully and comprehensively. 

5. Conclusion is elaborately rewriting based on the findings in terms of knowledge and practices of EBF.

---

## [Editor Report · Decision Letter 1]

7 Apr 2020

Knowledge and practices of exclusive breastfeeding among mothers in rural areas of Rajshahi district in Bangladesh: A community clinic based study

PONE-D-19-28733R1

Dear Dr. Islam,

We are pleased to inform you that your manuscript has been judged scientifically suitable for publication and will be formally accepted for publication once it complies with all outstanding technical requirements.

With kind regards,

Russell Kabir, PhD

Academic Editor

PLOS ONE
---

## [Editor Report · Acceptance letter]

14 Apr 2020

PONE-D-19-28733R1 

Knowledge and practices of exclusive breastfeeding among mothers in rural areas of Rajshahi district in Bangladesh: A community clinic based study 

Dear Dr. Islam:

I am pleased to inform you that your manuscript has been deemed suitable for publication in PLOS ONE. Congratulations! Your manuscript is now with our production department. 

With kind regards,

on behalf of

Dr. Russell Kabir 

Academic Editor

PLOS ONE